# Effects and Prospects of the Vibration Isolation Methods for an Atomic Interference Gravimeter

**DOI:** 10.3390/s22020583

**Published:** 2022-01-13

**Authors:** Wenbin Gong, An Li, Chunfu Huang, Hao Che, Chengxu Feng, Fangjun Qin

**Affiliations:** 1College of Electrical Engineering, Naval University of Engineering, Wuhan 430033, China; herwbin@yahoo.com (W.G.); lian196101@126.com (A.L.); wateriness_e@163.com (C.H.); hg15441@126.com (H.C.); 2College of Weapon Engineering, Naval University of Engineering, Wuhan 430033, China; kerryfengcx@126.com

**Keywords:** atomic interference gravimeter, active vibration isolation, vibration compensation, vibration noise, gravity measurement

## Abstract

An atomic interference gravimeter (AIG) is of great value in underwater aided navigation, but one of the constraints on its accuracy is vibration noise. For this reason, technology must be developed for its vibration isolation. Up to now, three methods have mainly been employed to suppress the vibration noise of an AIG, including passive vibration isolation, active vibration isolation and vibration compensation. This paper presents a study on how vibration noise affects the measurement of an AIG, a review of the research findings regarding the reduction of its vibration, and the prospective development of vibration isolation technology for an AIG. Along with the development of small and movable AIGs, vibration isolation technology will be better adapted to the challenging environment and be strongly resistant to disturbance in the future.

## 1. Introduction

Gravity is the force applied by the attraction of the Earth to the objects on the ground. Under the effect of gravity, an object descends at an accelerated speed, so that its acceleration is called gravitational acceleration. Gravitational acceleration varies with time and space. At a place on the Earth, gravitational acceleration is dependent on such factors as local latitude, altitude, landform, density and distribution of underground matters. As an important parameter in describing the gravitational field of the earth, gravitational acceleration has been extensively applied in studies of inertial navigation, geological survey, geophysics and basic physics, etc. [1,2,3]. Today, high-precision gravity field maps, like remote sensing satellite images, SAR images and other satellite images, play an important role in the field of national economy and people’s livelihood [4,5,6,7]. In the underwater navigation of submarines, the traditional sonar technology is unable to meet the requirements for high precision navigation at the seabed since it cannot receive any signal in deep waters, but a seabed high precision gravity map can be used to assist submarines in rapidly locating and avoiding the obstacles at seabed [1,8,9]. Hence, underwater gravity navigation entails high precision gravity measurement.

De Broglie claimed that physical particles had a wave particle duality. Like lights, atoms could be interfered by laser in beam splitting, reflection and combination because of their wave nature. In 1991, Kasevich and Chu et al. [10] utilized stimulated Raman transition in the coherent manipulation of cold atomic cloud for the first time, and implemented a cold atomic interferometer. Based on the measured gravity, the averaged resolution of the interferometer for 1000 s was 3 × 10^−6^ g. In 1992, Kasevich and Chu et al. [11] designed the world’s first atomic fountain gravimeter based on the stimulated Raman transition, and achieved the gravity measurement resolution of 30 uGal ($1\mathrm{uGal}=10^{-8}m/s^{2}\approx 10^{-9}g$) within the integral time of 2000 s. Continual improvements were then made for eliminating the local gravity error driven by earth tides, so as to achieve the measurement resolution of 0.3 uGal within the integral time of 60 s, and realize more accurate gravitational acceleration measurement [12].

An atomic interference gravimeter (AIG), as a high precision measuring device for absolute gravity based on atomic interferometer, can be used to measure the absolute value of gravitational acceleration. At present, more than 50 research and development institutions are engaging in the exploration worldwide including LNE-SYRTE in France [13,14,15], Humboldt University in Germany [16,17], Stanford University in USA [18], Zhejiang University of Technology (ZJUT) [19,20], Huazhong University of Science and Technology (HUST) [21,22], University of Science and Technology of China (USTC) [23], and Wuhan Institute of Physics and Mathematics (WIPM) [24,25] in China, among others [26,27,28,29,30,31]. A short list is given in Table 1. In 2009, LNE-SYRTE in France developed the world’s first movable atomic gravimeter [15]. While improving the sensitivity of atomic gravimeter, more efforts must be made to develop the atomic interference gravimeters of small size and good mobility. For this purpose, a number of dynamic experiments have been carried out ever since [20,32,33,34,35,36,37,38].

Compared with the traditional absolute gravimeter based on laser interferometer, an atomic gravimeter uses cold atomic cloud as a material to measure gravity. It can measure the gravity continuously for a long time and without mechanical wear, and achieve the sensitivity and accuracy as good as an FG5 gravimeter. Presently, the best combined standard uncertainty of atomic gravimeters can be up to 4.5 uGal in the world [46]. Due to its high measurement sensitivity and accuracy, an AIG has been widely applied in the accurate measurement of physical quantities including gravitational acceleration [37,47], gravity gradient [48], and universal gravitational constant [26]. Therefore, it is of great value in underwater navigation, resource scanning, and geologic monitoring, etc. [48,49,50,51].

The noise source of AIG is mainly composed of detection noise, vibration noise, Raman optical phase noise, optical frequency shift noise, etc. [52]. With the prior art, the detection noise, phase noise and frequency noise can be reduced to the level of mrad/single measurement; however, the vibration and noise can only be reduced to the level of 10–100 mrad/single measurement even if complex active and passive damping platforms are used, which makes the vibration and noise become the main noise source limiting the sensitivity of AIG.

Vibration, as a common phenomenon in the nature, often undermines the stability and reliability of various engineering instruments or equipment. Hence, vibration isolation is indispensable for these instruments and equipment [53,54]. In most engineering applications, attention has to be mainly paid to the vibration noise at 10–100 Hz or higher frequency, e.g., various means of transport and engines. The vibration at these frequency bands can be satisfactorily suppressed by simply applying the passive vibration technology. Nevertheless, more attention is paid to the vibration noise at low frequency and even very low frequency in some precision measurement experiments, e.g., microscopes, laser frequency stabilization systems and gravitational wave detection experiments [55,56,57]. For the purpose of higher measurement sensitivity, the good technology of low frequency vibration isolation should be adopted in these experiments [58,59,60]. When an AIG is used to measure gravitational acceleration, its measurement accuracy is dependent on a variety of factors with the increasing accuracy and the decreasing scale of measurement. Among these factors, the vibration of a Raman light reflection mirror significantly affects the measurement accuracy of an AIG, while vibration noise becomes a considerable restriction over its measurement accuracy and reliability [61,62]. Consequently, the technology for isolation and attenuation of vibration noise is crucial to accurately obtaining the atomic interference phase and implementing the accurate information detection of gravitational field [63,64].

The measurement error caused by Raman mirror vibration in the process of atomic interference can be effectively suppressed by differential measurement, such as atomic interference gradiometer [65]. Two sets of interference platforms at different positions in the atomic interference gravity gradiometer share the same optical and electronic system, and the two groups of atoms share the same pair of Raman light in the interference process, which can better suppress the Raman phase noise, vibration noise and other common mode noise in the measurement process without vibration isolation system, so as to obtain higher performance indexes than a single gravimeter. However, the measuring principle of AIG is different from that of gradiometer. There is only one interference platform, it is difficult to suppress or eliminate the vibration noise measured by a single atomic interferometer from the change of interferometer configuration, and can only be controlled from the vibration source or propagation path.

According to the noise transfer function of AIG, the low-frequency vibration noise has a great impact on AIG, while the high-frequency noise is attenuated by the mirror vibration noise transfer function. The response to the AIG to vibration and noise is like a low-pass filter. The cut-off frequency is the reciprocal of the time interval of the Raman light pulse of the atomic interferometer. The low-frequency vibration and noise has the greatest impact on the measurement. High frequency vibration and noise can be isolated by passive vibration isolation platform. For low-frequency vibration, the passive vibration isolation platform has no or little effect. In order to further improve the measurement accuracy of AIG, restraining low-frequency vibration and noise is the main research direction.

The other sections of this paper are organized as follows: the second section gives an introduction to the working principles of an AIG; the third section presents an analysis on how vibration noise affects the error of an AIG; the fourth section contains a review of the research findings regarding the vibration isolation technology for an AIG; and the fifth section gives a summary and discussion of the prospective development of the vibration isolation technology for an AIG.

## 2. Working Principles of an AIG

Like light wave interference, matter wave interference needs a beam splitter and combiner, which are often implemented by virtue of two-photon stimulated Raman transition in an atomic interferometer [10]. An AIG applies three Raman pulse beams onto an atomic wave packet to implement beam splitting, reflection and combination, so as to achieve atomic interference. Atoms are affected by gravity in the interference process. The gravity information can be obtained from atomic interference fringe to implement gravity measurement in the principles as shown in Figure 1a.

Under normal temperature, atoms are not easily manipulated since they move fast. Therefore, atoms must be slowed down first to generate cold atomic cloud. The dipole force and scattering force under the interaction of light and atoms are utilized for laser cooling and trapping of atoms in the vacuum cavity [66]. Rubidium atoms are decelerated by the Doppler cooling effect of resonant laser to generate cold atomic cloud in the end. Polarization gradient cooling is further implemented using the magnetic field generated by the Helmholtz coil in a magneto-optical trap (MOT) and a light field generated by three pairs of circular polarization lasers with propagation directions vertical to each other. This brings the temperature of cold atomic cloud to the uK level [67]. The decelerated cold atomic cloud is eventually trapped in the MOT as shown in Figure 1b. After shutting down the magnetic field, state preparation is conducted with the microwave state selection method when atoms move downward freely under the effect of gravity. The atoms at the level of basic state, which are sensitive to the magnetic field and have zero magnetic quantum in the atomic cloud, are therefore selected [68].

As shown in Figure 1c, MOT is utilized to prepare a cold atomic cloud. A sequence of three Raman pulse beams (π/2−π−π/2) is then used and applied to cold atoms for the beam splitting, reflection and combination of the atomic wave packet. Meanwhile, chirped scanning is carried out for the frequency of Raman transition to regulate the interference phase. It is assumed that atoms are in the state |a〉 at the initial time. After interaction with the first pulse beam π/2, atoms are in the superposition state of |a〉 and |b〉 have different momentums, and their difference is hk→eff. The atomic wave packet experiences beam splitting, and then evolves along two paths in space. In the meanwhile, the phase of light field is also transferred to atoms. After the free evolution time T, the second pulse beam π interacts with atoms. At this time, the atoms in the state |a〉 transit to the state |b〉, while the atoms in the state |b〉 transit to the state |a〉. The momentum of atoms also changes correspondingly, causing the reorientation of the atomic wave packet. After the time T, the third pulse beam π/2 interacts with atoms. Under this circumstance, beam combination happens to the atomic wave packet, causing the interference.

In the detection area, the time of flight method (TOF) is employed to detect the fluorescence signal of atoms, and obtain the normalized atomic transition probability P of atoms in two states [33,69,70]. In this case, P represents the probability of atomic interference in the macroscopic observable, and can be described by the following function:(1)$$P=P_{0}+C\cdot \cos(\Delta \Phi )/2$$
where $P_{0}$ is the average of atomic transition probability; C is the contrast of atomic interference fringe; ΔΦ is the measured phase difference of interference fringe, which is originated from the accumulation of atomic interference in different paths, and expressed as
(2)$$\Delta \Phi =\phi_{1}-2\phi_{2}+\phi_{2}=(k_{eff}g-2\pi \alpha )T^{2}$$
where $\phi_{1},\phi_{2},\phi_{3}$ is the Raman light phase at the time of interaction with three Raman pulses, respectively; α is the chirp rate of Raman laser, which is used to compensate for the gravity-induced Doppler frequency shift by linearly scanning the Raman light frequency difference; $k_{eff}$ is the equivalent wave vector; T is the time interval between two adjacent Raman light pulse beams, i.e., free evolution time of atoms.

Doppler frequency shift is positively correlated with the square of fall time [71]. In the implementation of an atomic interferometer, the Doppler frequency shift caused in the fall of atoms should be compensated to ensure the resonance of atoms with all three Raman pulse beams. The Raman light frequency difference is linearly scanned to compensate for the Doppler frequency shift. When the Doppler frequency shift can be perfectly compensated, cold atomic cloud always resonates with Raman light. The phase difference is $\Delta \Phi =(k_{eff}g-2\pi \alpha )T^{2}=0$. The gravitational acceleration measured by a cold AIG is $g=2\pi \alpha /k_{eff}$.

## 3. Influence of Vibration Noise on an AIG

The noise of AIG is mainly composed of detection noise, vibration noise, Raman optical phase noise, optical frequency shift noise, etc. With the prior art, the detection noise, phase noise and frequency noise can be reduced to the level of mrad/single measurement. However, the vibration and noise can only be reduced to the level of 10–100 mrad/single measurement even if complex active and passive damping platforms are used, which makes the vibration and noise become the main noise source limiting the sensitivity of cold atom gravimeter.

An AIG is presently one of the highest precision measuring devices for gravitational acceleration g in the world. In the measurement experiment, external vibration noise is very easily coupled into the total phase of inference fringe by virtue of the vibration of the Raman light reflection mirror in an atomic interferometer. This results in some measurement error. The vibration noise of most optical components in an atomic interferometer is in common mode during measurement. Therefore, phase noise can be caused to interference fringe only by the vertical vibration displacement of the Raman light reflection mirror placed at the bottom of the interferometer, and then undermine the sensitivity of the instrument [72].

The concept of sensitivity function is normally taken to describe the time domain atomic interferometer [63,73]. It is assumed that Raman light phase ϕ jumps δϕ at the time t, causing the variation δP(δϕ,t) of transition probability P. The sensitivity function $g_{s}(t)$ may be defined by
(3)$$g_{s}(t)=2\underset{\delta \phi \to 0}{\lim}\frac{\delta P(\delta \phi ,t)}{\delta \phi }$$

Interference signal has an approximately linear relationship with phase. The relationship between phase change and variation of transition probability is $\sigma_{\Delta \phi }=2\sigma_{P}$. In terms of phase change, the sensitivity function is therefore expressed as
(4)$$g_{s}(t)=\underset{\delta \phi \to 0}{\lim}\frac{\delta \Phi (\delta \phi ,t)}{\delta \phi }$$

The variation of transition probability is solved by segmenting evolution matrix. The time at the center of the second pulse beam π is taken as the time origin to obtain the sensitivity function of atomic transition signal to Raman light as follows:(5)$$g(t)=\{\begin{array}{cc} \sin\Omega_{R}(T+t) & -T\leq t<-T+\tau \\ 1 & -T+\tau \leq t<-\tau \\ -\sin\Omega_{R}t & -\tau \leq t<\tau \\ -1 & \tau \leq t<T-\tau \\ -\sin\Omega_{R}(T-t) & T-\tau \leq t<T \end{array}$$
where $\Omega_{R}$ is the Rabi frequency; τ is the length of finite Raman pulse; and T is the Raman pulse time interval. Transfer function is a weighting function for the relationship between system input and output. It is the Fourier transform of the sensitivity function $G\left(\omega \right)=\int_{-\infty }^{\infty }g_{s}(t)e^{i\omega t}dt$. The transfer function between the interference phase of an interferometer and the Raman light modulated phase is $H_{\phi }(\omega )=\omega G(\omega )$.

In the measurement of atomic interference gravity, a reflection mirror is used to generate a pair of Raman lights in opposite directions, which jointly affect the atomic cloud. The effective laser phase perceived by atoms is originated from the phase difference between two Raman light beams that are transmitted downward and reflected by the reflection mirror, respectively. Hence, the change of laser phase is directly attributed to the motion of the Raman light reflection mirror at the stage of interference. Noise is then introduced. It is assumed that two Raman light beams in an atomic interferometer have the wave vector $k_{1},k_{2}$, respectively. If the reflection mirror has the vertical vibration displacement $\delta_{z}(t)$, the phase noise introduced by the Raman light because of vibration is $k_{eff}\delta_{z}(t)$. The phase difference of interference fringe $\Delta \Phi =\phi_{1}-2\phi_{2}+\phi_{3}+k_{eff}\delta_{z}(t)$. When the vibration noise $k_{eff}\delta_{z}(t)$ is large enough to make the influence of vibration on ΔΦ greater than π, the interference fringe of atoms will be entirely eliminated. It is therefore evident that the phase of an AIG is affected by the position change of the reflection mirror. Vibration isolation should be therefore provided for the reflection mirror to achieve more accurate measurement of gravitational acceleration. The power spectral density of the Raman reflection mirror phase can be expressed as
(6)$$s_{\phi }(\omega )=k_{eff}^{2}s_{a}(\omega )/\omega^{4}$$
where ω is the angular frequency of vibration; and $s_{a}(\omega )$ is the noise power spectrum of vibration acceleration. The phase variance of an AIG can be expressed as
(7)$$\sigma_{\phi }^{2}=\int_{0}^{\infty }H_{\phi }^{2}(\omega )k_{eff}^{2}s_{a}(\omega )/\omega^{4}d\omega$$

The influence of the reflection mirror vibration noise on gravity is defined by
(8)$$\sigma_{g}^{2}=\int_{0}^{\infty }H_{a}^{2}(\omega )s_{a}(\omega )d\omega$$
where $H_{a}^{2}(\omega )$ is the transfer function of the reflection mirror vibration noise to the gravimeter. While satisfying $\omega \ll \Omega_{R}$ and τ≪T, there is
(9)$${\left|H_{a}(\omega )\right|}^{2}=\frac{k_{eff}^{2}}{\omega^{4}}{\left|H_{\phi }(\omega )\right|}^{2}=\frac{16\sin^{4}(\omega T/2)}{T^{4}\omega^{4}}$$

The transfer function of the reflection mirror vibration noise for an AIG can be used to create the transfer function curve of the vibration noise at different Raman light time intervals T as shown in Figure 2. As shown in the figure, when interrogation time T is 50 ms, 90 ms, 120 ms, respectively, the noises with the frequency lower than 1 Hz are all transferred nearly at the ratio of 1:1 to the interference phase. When the frequency is greater than 1 Hz, the transfer rate of vibration begins to attenuate. Additionally, when the frequency exceeds 10 Hz, the transfer rate has attenuated by five orders of magnitude. Evidently, the vibration noises with the frequency lower than f=1/T has the highest influence on measurement. The increasing frequency causes the transfer rate of vibration to attenuate at the rate $f^{2}$. Hence, the transfer function of vibration noise is typical of low pass just as a low-pass filter. In other words, it is more sensitive to low frequency vibration. The vibration noises with the frequency lower than 10 Hz affect a cold atomic gravimeter most significantly. For this reason, special attention must be paid to the vibration of low frequency bands for the vibration isolation system of an atomic gravimeter.

## 4. Research Status of Vibration Isolation Technology for an AIG

Vibration noise may be reduced by causing the attenuation of vibration and lowering the motion of the reflection mirror as much as possible. This can be implemented in a low noise environment, e.g., cold atomic gravimeter [13] (CAG), which had been used for gravity measurement in the Walferdange Underground Laboratory for Geodynamics. Nevertheless, a gravimeter will be significantly limited to a laboratory and not applied extensively if the measurement of gravitational acceleration is performed only in a low noise environment [74].

Vibration isolation is one of the major methods for vibration control. In this method, a vibration isolation system is used to isolate a vibration source from precision instruments. At present, vibration noise is mainly suppressed in three ways, including passive vibration isolation, active vibration isolation and vibration compensation. Among them, the last two ways are mainly applied for an AIG.

### 4.1. Passive Vibration Isolation

A passive vibration isolation system relies on an elastic damping material and mechanical structure to absorb or attenuate the mechanical waves of vibration. It is normally a mass-spring-damping system as shown in Figure 3. The attenuation of vibration is mainly achieved by such devices as a coil spring, elastomer pad, and air spring. The advantage of this system is the realization of the best vibration isolation with a simple structure, but not relying on any external energy, sensor, actuator or control system. Nevertheless, this system is troubled by very poor isolation of vibration at low frequency bands, very long time needed for stabilization, and low operability. It is often applied in industrial equipment, civil engineering structure, precision instrument and equipment. The common devices of passive vibration isolation include pneumatic vibration isolator [75], zero-length spring vibration isolator [76], and negative stiffness spring vibration isolator [77,78,79].

A zero-length spring based on long period is one of the main passive vibration isolation system designs for an AIG, IMGC-02 absolute gravimeter adopts this passive vibration isolation method [80,81]. It mainly consists of zero-length spring structure, geometric reverse spring structure, Euler column spring structure, and torsion balance spring structure, etc. In [76], Li et al., of Tsinghua University utilized feedback control to improve the zero-initial-length spring structure, devised and manufactured an ultralow frequency vertical vibration isolator based on spring link. The vibration isolation system employed the optical lever to detect the angular displacement of swing link. The feedback circuit controlled the voice coil motor to drive the swing link based on the detected displacement signal, which compensated for the influence of creep and temperature shift on the spring. After careful modulation, the system could achieve the stable oscillation within the natural cycle of 32 s, and constantly operate for more than one year. The system had been tested in T^−1^ absolute gravimeter, and realized the uncertainty of 2 μGal in 12 h measurement. At present, compared with other types of passive vibration isolation structures, zero-length spring structure is widely used.

However, because the spring needs a large volume, the volume of the zero-length spring structure is large, and will continue to accumulate due to the influence of temperature drift and creep. The requirements of miniaturization and mobility are difficult to meet the requirements of vibration isolation.

The negative stiffness spring can locally reduce the overall stiffness of the spring. The greater the spring stiffness, the stronger the bearing capacity and the greater the natural frequency. The negative stiffness spring is connected in parallel with the positive and negative stiffness springs, so that the whole has nonlinear characteristics near the equilibrium position, and the stiffness is close to 0. Negative stiffness vibration isolation system is widely used in gravity measurement experiment based on atomic interference, which can realize low vibration environment [79].

The research and development cost of high-precision passive vibration isolation platform based on complex spring and support structure is high, while the passive vibration isolation platform based on negative stiffness is simple and efficient, and can provide multi degree of freedom vibration isolation. The negative rigid commercial vibration isolation platform developed by Minus K company has good vibration isolation performance, small volume, simple operation, and it is easy to retrofit and install a voice coil motor [32]. It is widely used in the vibration isolation of the reflector of atomic gravimeter with good effect. The 100BM-10 commercial passive vibration isolation platform produced by the company is only in size 310 mm×310 mm×117 mm, with a payload range of 34–50 kg. It can provide 0.5 Hz vertical natural frequency and 1.5 Hz horizontal natural frequency, which is suitable for miniaturized system applications.

The Müller research team of the University of California at Bernoulli [32] used the passive damping platform (25BM-10, Minus K) to carry out vehicle flow static gravity measurement, and obtained the measurement sensitivity of 500 $\mathrm{uGal}/\sqrt{\mathrm{Hz}}$ and the measurement accuracy of 40 μGal.

A passive vibration isolation system can bear very high loads regardless of its simple structure, but depends much on the assembly and debugging accuracy of structure. It is essentially dependent on the elastic components of special structure for good vibration isolation, so that it is easily affected by the creep and temperature shift of elastic materials. For this reason, the system cannot maintain its good vibration isolation for a long time, and has poor resistance to disturbance. Its performance of vibration isolation is normally inferior to that of an active vibration isolation system, so that the active isolation is more common.

The passive vibration isolation platform can suppress high-frequency vibration and noise, but it has a poor suppression effect on low-frequency vibration and noise below 0.5 Hz, and even resonance will occur and increase vibration and noise. According to the transfer function of vibration, low-frequency vibration noise has a greater impact on atomic interference than high-frequency noise. Therefore, the passive vibration isolation system cannot meet the high-precision requirements of atomic gravimeter.

### 4.2. Active Vibration Isolation

An active vibration isolation system is equipped with a vibration sensor and actuator. By virtue of feedback control, it can make an effective improvement to the poor vibration isolation of a passive vibration isolation system at low frequency bands. This system is normally consisted of spring, sensor (accelerometer or seismograph), actuator and control system [82], as shown in Figure 4. Vibration signal is converted into the output signal of a brake through amplifier and control circuit. The feedback control is imposed on the vibration isolation platform to effectively control vibration. As the active vibration isolation can use a certain control algorithm according to the vibration signal of the sensor and use feedforward or feedback to achieve the control effect, compared with the passive vibration isolation platform, it can produce lower resonance frequency and achieve a stronger effect of vibration suppression. An active vibration isolation system has been widely applied in metrological, photoetching and medical fields as well as semiconductor industry. In the study of the active vibration isolation system for an AIG, research institutions have developed a variety of vibration isolation systems and achieved good effective improvements to measurement accuracy and sensitivity [83,84,85].

Hensley et al. of the Stanford University, USA designed an experimental system with vertical ground motion and atomic gravimeter isolation as given in [86]. The system combined an active system with a passive system formed by a mechanical spring and an optical workstation suspended in the compressed air. The active system was used to measure the acceleration of an object to be isolated, and then fed it back to an electromagnetic actuator as an offset against the motion. Eventually, an active spring vibration isolation system was developed with the effective natural resonance frequency of 0.033 Hz. It could lower the vibration noise to 10^−8^ $g/\sqrt{\mathrm{Hz}}$ from 0.1–20 Hz. The vibration noise of the system was reduced by 300 times. The system was tested in an atomic interference measurement experiment to obtain the uncertainty 3 × 10^−9^ g. In the meanwhile, a comparative experiment was carried out to prove that acceleration error signal could be lowered by 30–1000 times when noise was at 10 Hz to 100 Hz, and by 1000 times when noise was above 100 Hz.

In [87], Freier of the Humboldt University of Berlin simplified the structure of active vibration isolation, and devised a single rate spring active vibration isolation system. The system followed the basic principle that an isolated platform was supported by a principal spring to isolate high frequency vibration noise and form a passive vibration isolation system. An accelerometer placed on the isolated platform was employed to detect the vibration of the platform. Through active feedback compensation, feedback force was then applied to compensate for the influence of such vibration, so as to achieve the equivalent ultra-long period and isolate the influence of low frequency vibration noise. The vibration noises within the range from 0.03 Hz to 5 Hz were suppressed by 200 times. The natural resonance frequency of the system was 0.025 Hz. The sensitivity of an atomic gravimeter reached 7 × 10^−8^ $g/\sqrt{\mathrm{Hz}}$.

Schmidt et al. developed a gravimetric atom interferometer (GAIN) as detailed in [88]. To reduce the mechanical vibration noise, a vibration isolation system with active feedback was devised by placing a reflection mirror on a retrofitted commercial passive vibration isolation platform (Minus K50BM-10). A feedback circuit was implemented by measuring the residual vibration of the vibration isolation platform with a Guralp CMG-3VL uniaxial force feedback accelerometer, so that a voice coil motor was used to feed back the vibration to a vibration isolator. An electronic feedback device was installed in a detached control unit to keep the small size of the sensor, which lowered the effective resonance frequency from 0.50 Hz to 0.025 Hz. Hauth et al. [16] further optimized the active vibration isolation system in GAIN given in [15], and achieved the same low resonance frequency for both horizontal and vertical axes by remodeling the passive vibration isolation platform on which the reflection mirror was placed. Additionally, an inclined workstation was also devised to obtain the atomic interference fringe with the pulse interval T = 230 ms and realize the measurement sensitivity of 3 × 10^−8^ $g/\sqrt{\mathrm{Hz}}$.

Following the vibration isolation scheme proposed by Schmidt, Zhou et al., of HUST made an improvement to the active vibration isolation system of a cold atomic gravimeter in 2012 as detailed in [22]. In this improved vibration isolation system, the uniaxial accelerator in the original design was replaced by a triaxial commercial seismograph (Guralp CMG-3ESP) to achieve the positive correlation of the system’s damping force with the output of sensor. In this way, the stability of the feedback system was further improved to better obtain the data of horizontal vibration. The improved system could suppress the vibration noise at 0.1–1 Hz by 100 times, and realize the sensitivity less than 1 × 10^−9^ $g/\sqrt{\mathrm{Hz}}$ to the vibration noise at lower than 2 Hz, and its natural resonance frequency 0.016 Hz. When the vibration isolation system was applied in an AIG for experimental gravity measurement, its sensitivity reached 5.5 × 10^−8^ $g/\sqrt{\mathrm{Hz}}$, and its resolution was 6.5 × 10^−9^ g within the integral time of 60 s, which was comparable to that of the most advanced atomic gravimeter.

In [89], Zhou et al., designed an ultralow frequency active vibration isolator that was able to suppress the vibration noise in three-dimensional directions simultaneously. In the system, a passive vibration isolation platform suppressed the vibration noise on the ground in three directions. A triaxial microseismograph was introduced to detect the residual vibration noise on the passive vibration isolation platform, and then the vibration signal was converted into a control signal for a voice coil motor. Feedback force was applied in the vertical direction and two horizontal axial directions to further attenuate the vibration noise on the ground. This could also lower the vertical vibration acceleration caused by coupling with the horizontal vibration, so as to realize the optimal vibration isolation. After the vibration isolation system formed a feedback circuit, the equivalent resonance frequency in the vertical direction was 0.01 Hz, while the equivalent resonance frequency in the horizontal direction was 0.083 Hz. The vibration noise in the vertical direction was suppressed by approximately 50 times within the frequency range of 0.2 Hz to 2 Hz, but the vibration noise in the horizontal direction was suppressed by around 5 times. When there was not any active vibration isolation in the horizontal direction, the vibration noise was $1.8\times 10^{-9}\;g/\sqrt{\mathrm{Hz}}$. When active vibration isolation was added in the horizontal direction, the performance of vibration isolation was mainly restricted by the self-noise of the sensor and the electronic noise of electronic devices, but the vibration noise was lowered to $7.5\times 10^{-10}\;g/\sqrt{\mathrm{Hz}}$. Hence, the atomic interferometer achieved the sub-microgal level of gravity measurement accuracy.

As detailed in [90], Tang et al. of University of Chinese Academy of Sciences designed and implemented a compact and stable active low frequency vibration isolation system to improve the active vibration isolation system devised by Freier [87]. In the system, the vibration signals detected by a seismograph (CMG-3VL) were processed in a digital control system and then fed back to a voice coil motor, which could control and suppress the vibration of the passive vibration isolation platform. The natural frequency of the system decreased from 0.8 Hz to 0.015 Hz, so that the vibration in the vertical direction was effectively suppressed. The vibration noise at the frequency of around 1 Hz was attenuated to $1\times 10^{-9}\;g/\sqrt{\mathrm{Hz}}$. The vibration noise was reduced by 100 times on the whole. Therefore, its measurement accuracy was considerably improved.

Luo et al. of ZJUT developed a compact low frequency active vibration isolation system as presented in [91]. A sliding mode robust control system was utilized to process and feedback the vibration signals detected by a seismograph, while a voice coil motor was employed to control and eliminate the motion of the passive vibration isolation platform. Within the frequency range of 0.1–10 Hz, the sliding mode robust control system achieved the power spectral density of residual vibration noise 99.9% lower than that of the passive vibration isolation platform to the maximum, and 83.3% lower than that of the lead-lag compensation control method to the maximum. Apart from better performance of vibration isolation, it needed a setting of only three parameters, and offered a strong resistance to disturbance.

Chen et al. of USTC constructed an easily hauled three-dimensional active vibration isolation system for a movable atomic gravimeter as detailed in [92]. The system could effectively isolate the motion on the ground to enhance the measurement sensitivity of a movable atomic gravimeter. With a devised comprehensive feedback algorithm, it could isolate the vertical vibration on the ground by three orders of magnitude and the horizontal vibration on the ground by one order of magnitude. At the frequency of below 10 Hz to which an atomic gravimeter was sensitive, the vibration noise in the vertical direction was suppressed to 4.8 × 10^–9^ $g/\sqrt{\mathrm{Hz}}$, while the vibration noise in the horizontal direction was lowered to 2.3 × 10^–7^ $g/\sqrt{\mathrm{Hz}}$. The influence of vibration noise on the sensitivity of an interferometer was reduced to below 2 $\mathrm{uGal}/\sqrt{\mathrm{Hz}}$, which was two orders of magnitude lower than that of the interferometer without a vibration isolation system.

To measure the vibration noise of a reflection mirror in an atomic gravimeter, Zhang et al. [93] introduced an evaluation scheme for measuring the mirror vibration noise of an atomic gravimeter with a Michelson Raman laser (MIRL) interferometer. The MIRL interferometer was composed of the intrinsic Raman beam of the atomic gravimeter and a four-channel phase shift detector. The scheme presented an approach of using an improved “AI-MI-AI” three-interference system with a “three-cornered hat” to measure the contribution of mirror vibration to measurement instability. Restricted by the equivalence principle, the approach could not give the absolute vibration of the reflection mirror, but it was very simple, inexpensive, efficient and accurate to apply. Therefore, it offered another way to evaluate the contribution of vibration noise to measurement instability apart from commercial seismograph or accelerometer.

In [94], Zhou et al. put forward a cold atomic interference active vibration isolation system based on linear auto disturbance rejection control (LADRC) algorithm to implement the effective isolation for the low frequency noise at 0.1 Hz and below. The active vibration isolation system was compact and stable while offering low frequency and good performance. The LADRC controller involved two parts, i.e., extended state observer (ESO) and control law state equation. The ESO in the LADRC controller could directly observe the total disturbance of the system and make a timely compensation for such disturbance. As an active vibration isolation system combining a commercial passive vibration isolation system with an electronic feedback circuit, the system achieved the effective resonance frequency of 0.0152 Hz. As a spring-mass system that could generate nearly critical damping, it could significantly reduce the influence of frequency vibration at 0.1–10 Hz. Within the frequency range of 0.1–5 Hz, the system reduced the vertical vibration by 1000 times. Within the frequency range of 0.1–2 Hz, it could suppress the noise to 2 × 10^−9^ $g/\sqrt{\mathrm{Hz}}$. Meanwhile, the system had a stable oscillation and a natural period of 66 s. Its performance was better than that of a classic lag compensation filter. The system required the adjustment to only a few parameters, and could be applied by simply adjusting the feedback gain of system state error.

Regardless of good performance, an active vibration isolation system still has some limitations especially when an AIG is used in a noisy environment. A compact vibration isolation system may only cause the vibration to attenuate by several orders of magnitude. A large vibration attenuation system may achieve better vibration attenuation, but it weighs more than one ton, and is not easy to carry because of its large size. Moreover, it has a very complex structure. Additionally, the system is often limited to a very small dynamic range, and requires the adjustment and consideration of environmental conditions including temperature. At the time of resonance, the system could not attenuate or properly suppress, but actually increase noise.

### 4.3. Vibration Compensation

The cold atomic interference gravity measurement based on vibration compensation satisfies the urgent need for the integrated and small weight measuring system [95]. It is applicable to measuring absolute gravity in the field and on a movable platform. Vibration compensation is to measure vibrations at the same time AIG is performed, and then after the measurement adjust the AIG data to compensate for the vibrations. The advantage is that it is necessary only to measure the vibrations accurately, but not to mechanically correct for them [62,96].

As shown in Figure 5, the Raman light reflection mirror at the bottom of an AIG in a vibration compensation scheme is normally installed above a sensor (accelerometer or seismograph) to monitor the vibration of the platform and evaluate the vibration noise. The sensor should be installed as close to the reflection mirror as possible, and also accurately leveled. After passing an analog-to-digital converter and a digital filter, the longitudinal output signal of a sensor is approximate to the vibration signal of the reflection mirror. The sensitivity function convolving with the vibration acceleration is used to calculate the phase shift resulted from vibration within a measurement period. In the end, the phase shift is compensated in the phase-transition probability curve to reconstruct the fringe and obtain the actual gravitational acceleration.

In 2009, Merle et al. utilized a low noise seismograph to independently measure the vibration noise of a portable atomic gravimeter on the ground as detailed in [97]. After comparing two measurement schemes, i.e., fringe fitting and nonlinear locking, they found that the average phase of an interferometer could be determined in the phase measurement even if its phase noise exceeded 2 π. The scanned fringe of vibration noise was fitted when the interaction happened in a very short period (2T=100ms). The sensitivity of measurement at night reached 5.5 × 10^−8^ g. In the experiment, they explored the sensitivity limit of an atomic gravimeter without vibration isolation. This provided an idea for vibration compensation, so that it was of great significance to the study of portable atomic gravimeter.

The Le Gouët group of the Observatory of Paris designed the M-Z atomic interference in [15,98] using vibration compensation. The acceleration signal of a seismograph test device on a passive vibration isolation platform was utilized to calculate the phase shift of interference fringe, which was caused by vibration noise in different time periods. In the data processing, the phase noise caused by vibration noise was deducted. By virtue of vibration compensation, an atomic gravimeter achieved the sensitivity 1.4 × 10^−8^ g. Moreover, the group proposed to compensate the phase of a laser device using the vibration noise detected by the seismograph, so that the influence of vibration noise on measurement was reflected in the phase shift of interference fringe in the interaction of atoms and laser.

As described in [99], Barret et al. adopted the complementary working mode of “cold atomic gravimeter/accelerometer + classic accelerometer” and the vibration noise correction technology in the gravity measurement experiment in a parabolic flight microgravity environment. They analyzed the probability density of atomic interference signals to obtain the displacement $P_{0}$ and amplitude A of atomic interference fringe and solve the phase of atoms. Subsequently, an accelerometer fixed to the Raman reflection mirror was employed to measure the vibration acceleration and estimate the fringe period in which the phase of atoms fell into. The phase of atoms was then corrected. The corrected phase of atoms corresponded to the resultant acceleration of vehicle acceleration and vibration acceleration. Therefore, the vehicle acceleration could be directly determined when the vibration acceleration was known. This experiment had realized a dynamic gravitational acceleration measurement in a real sense for the first time.

Muquans, a French company, also successfully realized the real-time vibration compensation in its absolute quantum gravimeter (AQG) as presented in [35]. A highly sensitive accelerometer (Nanometrics Titan) was utilized to measure the vibration of a gravity measurement system. The acceleration data was then filtered and digitalized to compensate for the phase change caused by vibration, so that AQG could ensure the gravity measurement of high sensitivity even under the effect of severe vibration. In the experiment, the long-term stability of AQG could be lower than 1 μGal for absolute gravity measurement.

Lautier et al. applied the signal of a classic accelerometer in the real-time phase correction of an atomic gravimeter, so that it could operate with the best performance when there was not any isolation platform [100]. Moreover, it overcame the dead time problem in continuous measurement. Therefore, it was made ready for applications in geophysical and inertial navigation.

In [52], a rubidium AIG designed by Logan Latham Richardson of the University of Hannover was tested for low noise environment and simulated high noise environment in a vibration compensation experiment. A low noise ultra-wide-band seismograph (Trillium 240) was employed for vibration compensation. When the free evolution time was set to 78 ms, its short-term uncertainty increased from 4.4 × 10^−6^ $g/\sqrt{\mathrm{Hz}}$ to 9.2 × 10^−7^ $g/\sqrt{\mathrm{Hz}}$. In the meanwhile, a relatively high noise accelerometer (Nanometrics Titan) was used for vibration compensation. When the free evolution time was set to 10 ms, its short-term uncertainty was enhanced from 7.4 × 10^−3^ $g/\sqrt{\mathrm{Hz}}$ to 1.0 × 10^−4^ $g/\sqrt{\mathrm{Hz}}$.

In [62], a seismograph, a reflection mirror and a vibration source were fixed to a large vibration isolation platform to implement the calibration of transfer function. Subsequently, the Raman light reflection mirror was placed against the seismograph (CMG-3ESP-C) to generate a vibration signal. After filtering and integral compensation, the vibration signal was aggregated with the original interference fringe for correction. The gravitational acceleration was eventually obtained after correction. The correction was implemented in the following way: commercial software was used to reconstruct and simulate the waveform by inversing the transfer function of the seismograph to the reflection mirror. The Fourier transform was applied to the collected waveform. After correcting the spectrum, the inverse Fourier transfer was utilized to inverse back the time domain waveform. When the free evolution time was set to 60 ms, the resolution of the system could reach 32 μGal after 25.6 s integration. As effectively proved in an experiment, the passive vibration isolation platform could be used to suppress the vibration of most frequencies. The vibration isolation method could be applied in compensation for the vibration of large amplitude at given frequencies, and could achieve better vibration isolation than the passive vibration isolation platform when there was severe vibration in the environment.

At present, studies have been gradually conducted on applying an AIG in dynamic measurement. As presented in [33], Cheng et al., conducted the absolute gravity measurement of a ship in moored condition. A high precision accelerometer was placed below a Raman reflection mirror to measure the power spectrum of hull acceleration noise. After correction by vibration compensation, the sensitivity of gravity measurement reached $16.6\;\mathrm{mGal}/\sqrt{\mathrm{Hz}}$, and the resolution of gravity measurement within the integral time of 1000 s was 0.7 mGal. Li et al. [101] carried out a lake navigation test for an AIG. Based on the preliminary vibration isolation with an inertial stabilization platform, an accelerometer was employed to measure the vibration residual error of the gravimeter. After correction by vibration compensation, the measurement accuracy reached the mGal level.

The vibration compensation method is technically difficult to implement because of heavy computing workload. With regard to transfer function, it is easy to inverse the transfer function of a seismograph or an accelerometer, but difficult to determine the transfer function of the seismometer/accelerometer to the reflection mirror. In addition, the inverse Fourier transform is very complex for signal filtering, and very difficult to implement. The vibration compensation method may achieve the same effect as the passive vibration isolation method. Compared with the vibration isolation method based on mechanical structure, the vibration compensation method has a great advantage, that is, better realizing the high precision gravitational acceleration measurement when there is any strong external disturbance. Additionally, the vibration compensation method can satisfactorily optimize the vibration in all frequency bands, and is of great practical significance to measure the gravitational acceleration in the harsh field environment. The advantages and disadvantages of the vibration isolation system of the AIG are shown in Table 2.

## 5. Conclusions and Outlook

An AIG has developed into a significant tool for precision gravity measurement. Its measurement accuracy, sensitivity and applicability have improved. However, vibration isolation is one of its key technical problems in the development of an absolute gravimeter. The performance of vibration isolation system directly affects the measuring results and observation accuracy of an absolute gravimeter, even up to the mGal level. For this reason, it becomes a key and difficult point in the development of an absolute gravimeter. Up to now, an atomic gravimeter has been gradually expanded from static measurement in a laboratory to dynamic measurement in the field. For the development of small atomic gravimeters, vibration isolation system is urgently needed to eliminate vibration noise and enhance measurement accuracy.

The vibration of Raman mirror in AIG and the vibration of reference prism in falling angle cubic gravimeter are the reasons for their measurement errors, respectively. When active vibration isolation and passive vibration isolation methods are used, their vibration isolation principle is the same, that is, to suppress the vibration of Raman optical mirror/reference prism. However, the principle of vibration compensation is different. AIG compensates the phase, while the falling angle cube gravimeter compensates the interference trajectory. Due to the long research time of the vibration isolation method of a falling angle cubic gravimeter, the method theory is relatively mature, which can provide a reference for AIG vibration isolation.

Active vibration isolation is currently a mainstream vibration isolation method for a high precision AIG. It is employed by most atomic interference gravimeters that can achieve the accuracy up to 10^−9^ g. To apply the method, many parameters should be adjusted in the control system. However, an AIG can achieve very high measurement accuracy if these parameters are properly adjusted. The method is restricted by its complex system design and higher requirement for small size of gravimeter. Additionally, the noise of instruments in the active feedback system may introduce an error, which is a major contributor to inaccurate measurement. For the purpose of higher measurement accuracy, it is very important to improve the performance of sensor and satisfactorily process vibration signal.

In the vibration compensation method, compensation is directly made for the calculated interference fringe of vibration signal. When external vibration is more noticeable and there are many violent disturbances, the vibration compensation method can achieve better results than other vibration isolation methods. Presently, vibration compensation is mainly dependent on the limited sensitivity of an accelerometer used to measure vibration noise. It still needs further improvement to implement high accuracy measurement of absolute gravity. Moreover, the accelerometer and the transfer function between accelerometer and Raman light reflection mirror were not taken into account in the vibration compensation. The cold atomic interference gravity measurement based on vibration compensation satisfies the urgent need for an integrated and small gravity measurement system, and applies to the measurement of absolute gravity in the field and on a movable platform. It is the trend of future development.

## Figures and Tables

**Figure 1 sensors-22-00583-f001:**
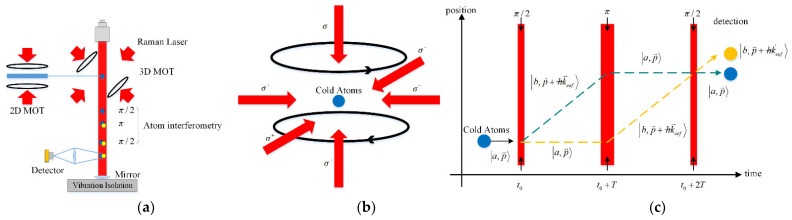
Principles of an AIG. (**a**) Operation diagram of an AIG; (**b**) Diagram of three-dimensional MOT atom cooling and trapping; (**c**) Diagram of Raman pulse atomic interference. Three Raman light pulse beams cause the beam splitting, reflection and combination of atomic wave packet to result in interference.

**Figure 2 sensors-22-00583-f002:**
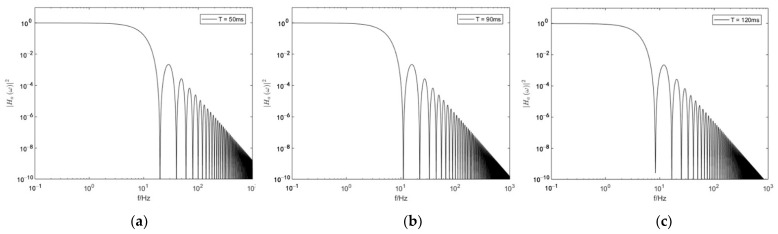
Transfer function spectrum of the reflection mirror vibration noise in an AIG at different time intervals under the effect of Raman light. (**a**) T=50 ms, (**b**) T=90 ms, and (**c**) T=120 ms.

**Figure 3 sensors-22-00583-f003:**
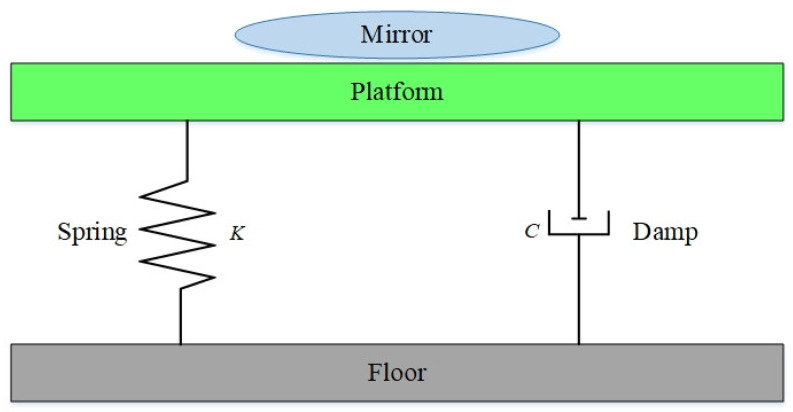
Composition of a passive vibration isolation system.

**Figure 4 sensors-22-00583-f004:**
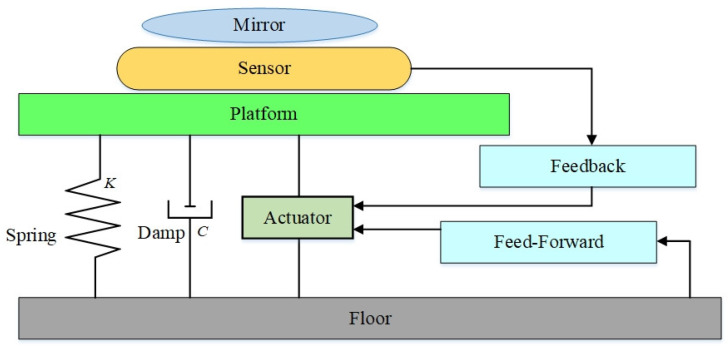
Composition of an active vibration isolation system.

**Figure 5 sensors-22-00583-f005:**
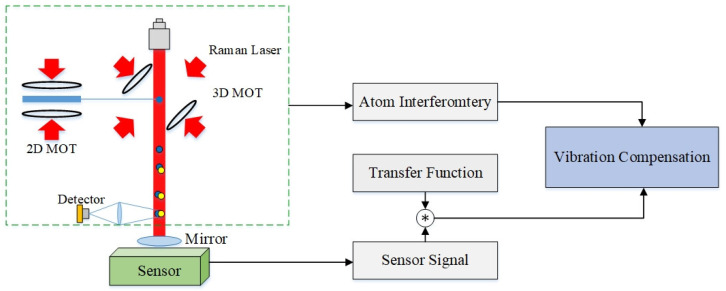
Vibration compensation scheme for an AIG.

**Table 1 sensors-22-00583-t001:** Overview of studies on AIG.

Year	Institution	Interrogation Time	Sensitivity	References
2001	Stanford	160 ms	$20\;\mathrm{uGal}/\sqrt{\mathrm{Hz}}$	Peters et al. [18]
2008	Stanford	400 ms	$8\;\mathrm{uGal}/\sqrt{\mathrm{Hz}}$	Holger Müller [39]
2019	University of California	130 ms	$37\;\mathrm{uGal}/\sqrt{\mathrm{Hz}}$	Wu et al. [32]
2008	LNE-SYRTE	50 ms	$1.4\;\mathrm{uGal}/\sqrt{\mathrm{Hz}}$	Le Gouët J et al. [15]
2013	ONERA	48 ms	$42\;\mathrm{uGal}/\sqrt{\mathrm{Hz}}$	Bidel et al. [36]
2018	Muquans	60 ms	$50\;\mathrm{uGal}/\sqrt{\mathrm{Hz}}$	Menoret et al. [35]
2014	LENS	160 ms	$30\;\mathrm{uGal}/\sqrt{\mathrm{Hz}}$	Rosi et al. [40]
2013	Humboldt University	260 s	$3\;\mathrm{uGal}/\sqrt{\mathrm{Hz}}$	Hauth et al. [16]
2016	Humboldt University	0.26 s	$9.6\;\mathrm{uGal}/\sqrt{\mathrm{Hz}}$	Freier et al. [17]
2013	ANU	60 ms	$60\;\mathrm{uGal}/\sqrt{\mathrm{Hz}}$	Altin et al. [41]
2013	HUST	300 ms	$4.2\;\mathrm{uGal}/\sqrt{\mathrm{Hz}}$	Hu et al. [42]
2019	HUST	160 ms	$53\;\mathrm{uGal}/\sqrt{\mathrm{Hz}}$	Luo et al. [43]
2014	Zhejiang University	60 ms	$10\;\mathrm{uGal}/\sqrt{\mathrm{Hz}}$	Wu et al. [44]
2018	National Institute of Metrology	70 ms	$44\;\mathrm{uGal}/\sqrt{\mathrm{Hz}}$	Wang et al. [31]
2019	ZJUT	120 ms	$300\;\mathrm{uGal}/\sqrt{\mathrm{Hz}}$	Fu et al. [19]
2019	WIPM	200 ms	$30\;\mathrm{uGal}/\sqrt{\mathrm{Hz}}$	Huang et al. [45]

**Table 2 sensors-22-00583-t002:** Comparison of vibration isolation systems for an AIG.

Methods	Previous Studies	Advantages	Disadvantages
Passive vibration isolation based	Li et al. [76] Refs. [80,81] Wu et al. [32]	It does not require external energy, sensor, actuator or control system, and can achieve the greatest vibration isolation with a simple structure.	It has very poor performance for the resonance and vibration isolation at low frequency bands, needs a very long stabilization time, and is poorly operable.
Active vibration isolation based	Hensley J. M. et al. [86] Freier [87] M. Schmidt et al. [88] M. Hauth et al. [16] Zhou et al. [22] Zhou et al. [89] Tang et al. [90] Luo et al. [91] Chen et al. [92] Zhang et al. [93] Zhou et al. [94]	It may generate lower resonance frequency, better vibration suppression effect, and realize the high precision of static measurement.	It has a complex system structural design, is difficult to eliminate own noise, and offers the low precision of dynamic measurement.
Vibration compensation based	S Merlet et al. [97] Refs. [15,98] Brynle Barrett et al. [99] Menoret, V. et al. [35] Lautier et al. [100] Logan Latham Richardson [52] Xu et al. [62] Cheng et al. [33] Li et al. [101]	It can effectively isolate vibration at all frequency bands, be strongly resistant to disturbance, adapt to the harsh field environment in applications, and meet the needs for small and movable atomic gravimeters in applications.	It causes a heavy computing workload, and is difficult to determine the transfer function between the measuring device and equipment, and to achieve high precision vibration isolation.

## Data Availability

Not applicable.

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
