# Peer review of "Effects and Prospects of the Vibration Isolation Methods for an Atomic Interference Gravimeter"

_sensors, 2022, doi:10.3390/s22020583_

Round 1

Reviewer 1 Report

Gong et al present a review article on the application and requirements of vibration isolation in atom interferometry. This is a useful survey of the existing literature, but I have some suggestions for improvements.

In Table 1, the third entry has no units for the sensitivity, and the implied units are not obvious. It would also be useful to convert the second entry to units of ugal/rt Hz so that all the data can be compared. Finally, the interrogation time of the interferometer is very relevant here, since that determines the bandwidth required for the vibration isolation. It would be useful to include it. (If room is an issue, I don’t know that the Country entry is that important, it is easy to infer from the Institution.) My impression is that the basic challenge here is that the AIG sensitivity and vibration isolation systems both act as low pass filters, so there is always a challenge of isolating sufficiently low frequencies. If the authors agree, it would be useful to explain this challenge up front, in general terms.

I think the introduction should mention gravity gradiometry as an alternative method to avoid vibrational sensitivity. Obviously this is not the same as absolute gravimetry, but it can be applied in many of the same circumstances.

I think the introduction should also say a little more about the bandwidth over which AIG measurements are sensitive. The paragraph starting at line 79 implies that AIG requires very low frequency isolation, but make this more explicit.

It would be interesting and useful to comment on the difference between AIG and falling corner cube gravimeters, in terms of their vibration sensitivity. I would expect them to be fairly similar but perhaps not.

Eq (4) at line 181 is confusing, δÏ• seems to be used in two different ways? Perhaps the notation can be improved or explained better.

At line 214, the comment that “noises with the frequency lower than 1 Hz are transferred...” would be more accurate and more informative if the correct corner frequency were stated in terms of the interrogation time T.

Also in Fig 2, is the assumption that ω  â‰ª Ω_R valid even at the relatively high frequencies shown? I worry that the plotted high frequency behavior is potentially misleading.

It would be quite helpful to give a basic explanation for why active isolation can provide lower frequency cutoffs than passive.

It is a little confusing at line 269 to say that a problem with zero-length springs is that they require a significant length. Maybe it is adequate to just say this method requires a large volume?

At line 301, it is confusing to say that passive isolation is “rarely applied in an absolute gravimeter,” given that the authors just listed several notable cases where it was applied. Perhaps saying that active isolation is more common would work better.

I found the explanation of vibration compensation less clear than I think it should be. I think the first paragraph should make clear that the concept is to measure vibrations at the same time AIG is performed, and then after the measurement adjust the AIG data to compensate for the vibrations. I would also make the point that the advantage here is that it is necessary only to measure the vibrations accurately, but not to mechanically correct for them.

At line 542, is the figure of 60 s correct? I think 60 ms is more likely.

In general, the English of the article is noticeably imperfect, but it is usually clear enough to understand for a native speaker.

Overall I think the article succeeds in providing a review of the topic, and it will probably be reasonably accessible to those outside the AIG field. If the improvements mentioned can be incorporated, I would recommend for publication.

Reviewer 2 Report

This review article mainly introduces the working principle of the AIG, analyzes errors of the AIG induced by vibration noise, gives the review of vibration isolation technologies for AIGs, and then gives  a summary and discussion of the prospective development of the vibration isolation technologies for AIGs. The manuscript is well organized and comprehensively described. 

However, language should be improved, such as the listed points:

  1. Line 569, “to measuring” should be “to measure”;
  2. Table 2, “give play to stronger vibration suppression” ?
  3. Line 573, “have been constantly improving” should be “improved”;
  4. Line 598, in the sentence “the transfer function of accelerometer…”, should be “the accelerometer” or “accelerometers”;

Other language issues like those should be carefully improved. 
